# Benefits of Respite Services on the Psycho-Emotional State of Families of Children Admitted to Hospice Palliative Care Unit: Preliminary Study on Parents’ Perceptions

**DOI:** 10.3390/healthcare12070748

**Published:** 2024-03-29

**Authors:** Mihaela Hizanu (Dumitrache), Estera Boeriu, Sonia Tanasescu, Ada Balan, Licinia Andrada Oprisoni, Maria Valentina Popa, Cristian Gutu, Dan Dumitru Vulcanescu, Iulia Cristina Bagiu, Radu Vasile Bagiu, Tiberiu Liviu Dragomir, Casiana Boru, Cecilia Roberta Avram, Letiția Doina Duceac

**Affiliations:** 1Doctoral School of Biomedical Sciences, Faculty of Medicine and Pharmacy, “Dunărea de Jos” University of Galați, 47 Domnească Street, 800008 Galați, Romania; mh202@student.ugal.ro (M.H.); mb582@student.ugal.ro (M.V.P.); 2Department of Pediatrics, “Victor Babes” University of Medicine and Pharmacy, Eftimie Murgu Square 2, 300041 Timisoara, Romania; tanasescu.sonia@umft.ro (S.T.); carstea.ada@umft.ro (A.B.); oprisoni.licinia@umft.ro (L.A.O.); 3Department of Oncology and Hematology, “Louis Turcanu” Emergency Clinical Hospital for Children, Iosif Nemoianu Street 2, 300011 Timisoara, Romania; 4Clinical Medical Department, Faculty of Medicine and Pharmacy, “Dunărea de Jos” University of Galați, 47 Domnească Street, 800008 Galați, Romania; cristian.gutu@ugal.ro (C.G.); doina.duceac@ugal.ro (L.D.D.); 5Department of Microbiology, “Victor Babes” University of Medicine and Pharmacy, Eftimie Murgu Square 2, 300041 Timisoara, Romania; dan.vulcanescu@umft.ro (D.D.V.); bagiu.iulia@umft.ro (I.C.B.); 6Multidisciplinary Research Center on Antimicrobial Resistance (MULTI-REZ), Microbiology Department, “Victor Babes” University of Medicine and Pharmacy, Eftimie Murgu Square 2, 300041 Timisoara, Romania; 7Department of Hygiene, Preventive Medicine Study Center, “Victor Babes” University of Medicine and Pharmacy, Eftimie Murgu Sq. Nr.2, 300041 Timisoara, Romania; bagiu@umft.ro; 8Medical Semiology II Discipline, Internal Medicine Department, “Victor Babes” University of Medicine and Pharmacy, Eftimie Murgu Square 2, 300041 Timisoara, Romania; dragomir.tiberiu@umft.ro; 9Department of Medicine, “Vasile Goldis” University of Medicine and Pharmacy, 310414 Arad, Romania; boru.casiana@uvvg.ro; 10Department of Residential Training and Post-University Courses, “Vasile Goldis” Western University, 310414 Arad, Romania; avram.cecilia@uvvg.ro

**Keywords:** life-threatening illness, respite, palliative care, children, demographics, cancer disease, health workers

## Abstract

Background: In children’s palliative care, the term “respite” refers to a temporary break offered to primary caregivers of a child with a life-limiting illness. The aim of this study was to assess the perceptions of parents who have benefited from respite care services in the Lumina Association, Bacău hospice unit and the benefits it can bring in improving their psycho-emotional state. Methods: The study consisted of quantitative research involving 34 parents/caregivers who responded to a questionnaire with 26 questions, and qualitative research which involved the organization of a focus group with 12 parents who benefited from respite services. Results: The use of respite services was associated with a significant reduction of psycho-emotional distress on the part of primary caregivers; 91% of respondents said that this type of service reduces the level of psycho-emotional stress. Conclusions: All participants in the study confirmed that the most important benefit of respite is the time gained to care for family and health. The development of respite services could reduce the risk of emotional exhaustion and mental health problems.

## 1. Introduction

Caring for children with life-limiting conditions is often emotionally, physically, spiritually and financially demanding [1]. A life-limiting condition in pediatrics is a condition in which the diagnosis occurs before the age of 16, there is no curative treatment and there is a high probability that the child will die prematurely at a fairly young age (approximately 25 years) [2].

Children with such a diagnosis are often cared for within the family, and caregivers, whether parents, foster parents, guardians, grandparents or siblings, often need a “short break” or “time off” [3], as providing ongoing care can often be exhausting.

Supporting the patient’s family and friends is an integral part of palliative care. The onset of a serious and progressive illness often causes questions, anxiety and distress in the relatives of the person affected. In addition, in the palliative approach, relatives have a particularly important role: they participate in discussing their needs, they share their knowledge about the person, they care for the person in distress and they express their opinions and questions [4]. 

Family and loved ones sometimes need to be supported in their difficult journey in the face of serious illness affecting one of their own. During the period of palliative care and regardless of where they are provided (hospital, hospice unit, home), the sick person and their loved ones need to be able to spend time together. In addition, the fact that “the family has an important influence on a person’s physical and psychological balance” is an active element in the patient’s environment that can influence their vital functions. Research and clinical observations in the health field demonstrate the influence of the family on people’s health behaviors and the course of an illness. The family is the patient’s first and main support. Everyone has their own way of experiencing illness and the period of hospitalization. Requirements and needs are not the same. There is mutual affection between these people, which is why we consider that the patient and his entourage constitute one and the same unit that the caregiver must take care of [5]. 

Despite advances in medical care, augmented by developments in pharmaceuticals, there has been an increase in the number of children diagnosed with life-limiting illnesses requiring palliative care and requiring complex treatment and complex medical care at home [6].

However, social support for families caring for such children has not kept pace with these developments [7]. Respite care, a form of social support, first identified in the literature as an unmet need 20 years ago, continues to be problematic today [8].

Families with children who are not experiencing significant health problems can rely on support from relatives, friends, childcare centers and schools. Conversely, children with serious conditions may be unable to attend a regular nursery or school, requiring assistance from specialist medical teams or carers, thus limiting support options for their parents. Respite care has been identified as a vital component of care for children with life-limiting conditions and is often cited as having benefits for both the child receiving central care and their parents [9].

Neufeld et al. 2001 defined respite care as: The provision by appropriately trained individuals of care for children with life-limiting conditions, for a specified period of time, thus providing temporary relief to the usual care giver [10].

Respite care for families of children diagnosed with life-limiting illnesses has become a significant concern in the medical and social fields in recent decades. It has emerged as an obvious need as the emotional and physical impact of ongoing care for children with serious illnesses on their families has become better understood.

In general, there is no precise date for the emergence of respite care for these families because the evolution of this concept has been gradual and influenced by many factors, such as advances in medicine, changes in social policy, and changes in public perception of the needs of these families.

Although the concept may seem new to some, it is not a recent phenomenon; it emerged in the late 1960s with the deinstitutionalization movement. One of the fundamental tenets of this movement was the belief that the best place to care for a child with special needs was in the home and community. Families who have a child with a disability or a progressive chronic illness know well the level of commitment and intensity of care needed for their children. This level of commitment and care becomes an integral part of everyday life, a family routine, but the same commitment can also lead to stress as part of this routine, and parents can become accustomed to the lack of time for themselves [11]. 

It is essential to clarify the meaning and purpose of respite care. As Miller, 2022 states in his article, the term “respite care” has often been criticized because it may suggest that the child is perceived as a burden that parents need to be relieved of [12]. Many parents do not see their child as a burden, but for some, the term accurately reflects their needs. Treneman et al. (1997) describe respite care as: “caring jointly for a person with difficulties/disabilities, either at home or in a specialist center, to give the family a break from routine care” [13].

However, the concept of respite care was also defined by Judd in 1995 as a form of complementary and flexible care provided in the home or other appropriate setting with specialist nursing care, with the aim of giving parents or carers a break and respite [14]. 

In recent years, there has been an increasing effort to integrate palliative care for children adequately into the Romanian health system. However, according to the study by Pacurari N et al. in 2021, there are still significant barriers preventing the expansion of palliative care implementation for children. These include a lack of qualified staff in this field, insufficient financial resources, political instability in the country and migration of qualified medical staff, which remain significant barriers to the uniform development of these services nationwide [15].

One of the most recent assessments of the development of pediatric palliative care globally [16], published in 2020, places Romania among the category 3a countries with isolated palliative care services dedicated to children.

The study concludes that only 21 of the 113 countries analyzed provide access to pediatric palliative care services at a reasonable level, and therefore less than 10% of the world’s population under 20 (35% of the global population) have access to timely palliative care services. More than 778 million children (30.7%), or about one third of the world’s children, live in the 55 countries (including Romania) where dedicated palliative care ser-vices are isolated and rare.

In 2019, in Romania, according to the last annual report of the Romanian Association of Palliative Care (https://www.studiipaliative.ro/wp-content/uploads/2021/04/Raport-furnizori-ingrijiri-paliative-Romania-2019.pdf, accessed on 3 January 2024), there were five palliative care (continuous hospitalization) services for children, two of which were of the “hospice” type developed by non-governmental organizations (20 beds) and three services of 5 beds each in the pediatric oncology wards of public hospitals (15 beds). According to the same document, there were only three non-governmental organizations with palliative care services for children with specialized teams at home.

The “respite” service, new in Romania, is a service that allows primary caretakers to take a “break” and leave their child in the care of a substitute, through formal and informal means. This aims to reduce the stress and fatigue that comes with the ongoing provision of care for a child with progressive chronic conditions [3,10,17]. Respite can also provide an opportunity for the child to participate in socialization activities, engage in combined therapies and be valued.

The main focus of palliative care is on quality of life and respecting the wishes of the sick person. There is no break between curative and palliative care. Palliative care starts well before the terminal phase, sometimes while specific treatments are ongoing. Palliative care aims to relieve physical pain and other symptoms, but also to take into account the psychological, social and spiritual suffering of the patient and their loved ones [4]. As such, the aim of this study was to assess parents’ perceptions of the use of respite care and the benefits it can bring in improving their psycho-emotional state.

## 2. Materials and Methods

The methods used to obtain the impact of respite care on the psycho-emotional state of families of inpatients in the care service were the self-administered questionnaire and the focus group.

### 2.1. Study Design

A semi-structured interview guide was developed, based on the literature and the information sought. The guide included questions about experience with the service and views on palliative care service provision within the Lumina Association—Children’s Palliative Care Centre, Bacau, Romania, a non-governmental organization located in the North-East region. The respite program started in September 2017 and was accessed by children diagnosed with life-limiting illnesses from all over Romania. There are 14 continuous hospitalization beds—hospice type—where a multidisciplinary team provides medical, social, psycho-emotional and spiritual support to children diagnosed with life-limiting diseases and their families. All services provided by the organization are free of charge.

Given that Romania is at the beginning of the development of pediatric palliative care and that there are few and unevenly developed services, we may have a starting point for the development of an instrument to study parents’ perceptions of the benefits of appropriate respite care to prevent physical and psychological exhaustion in families caring for children diagnosed with life-limiting illnesses, whose care sometimes lasts for years.

Previous research, although very little [18,19], conducted on the need for respite for families caring for a child diagnosed with a pediatric life-limiting illness, formed the theoretical basis of this study, contributing to the identification of problems, the development of the hypothesis and questionnaire items and the overall direction of the research. The use of this theoretical framework facilitated the development of a scientifically grounded approach to investigating parents’ perceptions and the benefits that a respite program can bring to their psycho-emotional state.

A questionnaire with 26 questions was used, based on the needs of, and feedback given by, patients and families who received respite care services in our center. Both categories and subcategories of questions integrated into the questionnaire were analyzed to address in detail the concept of respite, the benefits and the need for this program. A special attention to the wording of the questions was given in order to make them as clear and relevant as possible for the respondents. Psychometric principles were followed to guarantee the validity and reliability of the questionnaire. Both multiple-choice and open-ended questions were chosen, in order to allow respondents to provide extensive and context-specific information.

These questionnaires provided data on the age, background, sex of respondents, respondents’ knowledge of the concept of “respite”, benefits of the “respite” program, effect of this type of service on the psycho-emotional state of the primary caregiver, age of the child and condition of the child. The questionnaires used have the advantage of collecting information from a large number of respondents at a relatively low cost, but also have the disadvantage of not being sufficiently in-depth. Furthermore, the information obtained through questionnaires is limited by the structure and content of the questionnaire itself. The data provided from the questionnaire were statistically analyzed and percentages were calculated. 

The quantitative research was conducted by applying a questionnaire comprising questions to one family member/caregiver who had cared for or spent the most time with the child, which was developed by the research team using information from the international literature [20,21]. The questionnaire is available as Appendix A.

### 2.2. Patient Selection

In selecting the study group, a non-probability sampling method was used, based on specific pre-determined criteria. By applying inclusion and exclusion criteria, we aimed to obtain a representative and adequate sample for our study. We also used the direct invitation sampling method, by contacting by telephone and visiting the homes of people who met the inclusion criteria to hand them the questionnaires. This method was effective in identifying and attracting people who met the selection criteria and who were interested and willing to participate in the study.

Study inclusion criteria were: (1) adult (>18 years) parents/family members who care for children diagnosed with a life-limiting illness; (2) caretakers with children who have received respite palliative care services at the Lumina Association—Children’s Palliative Care Centre.

Exclusion criteria: (1) participants who were not able to complete the questionnaire; (2) participants who could not provide relevant and valid information for this study.

Respondents were adults without cognitive impairment who consented to participate in this study. The quantitative research included the following: -Administration of the questionnaire by the researcher to a family member/caretaker;-Centralization of results and analysis of data provided.

### 2.3. Structure and Interpretation of Data

Each questionnaire contained two parts: in the first part, the respondent was asked to write down characteristics about the background and then to make judgements about the term “respite” and what the benefits of respite services might be. In the second part of the questionnaire, the respondent was asked to answer a few questions about the child they cared for, how much time they spent caring for the child and whether they had benefited from respite services. Also, in the second part of the questionnaire, respondents indicated the benefits of this respite program. The questions had multiple answers.

On the other hand, the qualitative research activity involved a focus group with 12 parents who had benefited from the respite service over a period of 14 days within the Palliative Care Centre, Bacau, Romania. 

In the focus groups, several methods to obtain and interpret the collected information were used [22]. These included included the following:

Moderation: Moderators (M.H. and L.D.D.), based on an interview guide used to facilitate the group discussions, guiding the topics and questions discussed and ensuring that all participants expressed their views and experiences.

Open questions and challenges: Open questions were used to stimulate discussion and allow participants to freely express their perspectives. 

Non-verbal observation: During the focus group sessions, the moderator was attentive to participants’ non-verbal language, such as facial expressions and body language, to better understand their reactions and emotions.

Audio/video recording: To ensure maximum accuracy in transcribing information, and with the consent of the focus group participants, audio recording equipment was used to record the focus group session. This helped later to reconstruct the discussions and accurately transcribe the information.

Data transcription and analysis: After the focus group sessions, the recordings were manually transcribed and relevant non-verbal observations were incorporated where appropriate. This was handled by two examiners (E.B. and S.T.).

Relevant information was grouped into themes and sub-themes in order to gain relevant and in-depth insights into the topic of interest and parents’ perceptions (for example: theme “time”).

The focus group aimed to assess the needs of the family and the impact of the child’s chronic progressive illness on the psycho-emotional state, and included the following: -Contacting participants—from the home care service database (for those who had not previously received respite care) and from the respite service database (for participants who had received at least two periods of respite care within 3 years);-Preparation of the interview guide, taking into account some aspects (families’ needs, their children’s quality of life and the influence of illness and child care on psycho-emotional health and family dynamics);-Preparing the secure meeting space;-Recording of the meeting.

The researcher’s discussion focused on the approach and the importance that the family exerted in influencing the physical and psychological balance of a person, and that the psycho-emotionally balanced family was an active element in the patient’s environment that could influence his vital functions.

The study was conducted in accord with the ethical standards of the Lumina Association—Children’s Palliative Care Center, Bacau and with the express agreement of the Ethics Commission, under decision number 1027/29.12.2023.

## 3. Results

### 3.1. Analysis of Questionnaires

A total of 34 caregivers—female family members who are the primary caregiver and spend the most time with and caring for the sick child and who received respite care services at the Palliative Care Center for Children, Bacau—were included in our study; they were visited at their home by the researcher and asked to complete questionnaires. 

Centralizing the results allowed us to find out what the most important benefits of respite care were to caretakers and to families, and how they found out about respite care. 

The “time” response associated with the main benefits is an important index that best assesses the impact of caring for children diagnosed with incurable diseases in specialized services on the psycho-emotional state of the family as well as that of the caretakers, and the need to develop and meet the needs of these families.

#### 3.1.1. Analysis of Demographic Data in the Study Group Included: Gender of Caregivers, Age and Background, Age of the Child and Type of Pathology the Child Suffers from

(a)Demographics of the group included in the study

The group of respondents who participated in the study and responded to the questionnaire used was 100% women, suggesting that the involvement in caring for sick children lies exclusively with the mother, who is most affected by the burden of care and who needs real support to cope with the multiple needs that arise in this journey with illness.

(b)Distribution of respondents by background

Figure 1 shows that 62% of respondents are from rural areas, with limited access to health services, the internet, information about specialist services or therapies for recovery and support through support groups, and 38% are from urban areas, indicating that both urban and rural areas have a similar burden of care and that caretakers need support and respite to cope with the challenges of the incurable disease.

The expectation that parents provide home care for children with complex needs is a recent phenomenon, as until recently much of the care these children needed would have been considered the domain of health professionals. The provision of in-home care has a significant impact on family life, with disruptions to sleep and restrictions in social activities [23]. 

Families who have a child with special care needs can become marginalized, leading to isolation [24], due to a lack of energy to carry out social activities, or because friends fear being asked to care for their child with complex needs.

(c)Distribution of respondents by age group

It can be seen in Figure 2 that in the category of respondents involved in caring for a child with a chronic progressive disease are predominantly people in the 30–39 age category, then those aged 40–49. Women at these ages (30–39 years), in full physical and intellectual capacity, often take on this responsibility alone, 24/24 h, in addition to household tasks or caring for other children; this situation has a major social impact, and can lead to depression, irritability and anxiety due to sleepless nights, household chores that often exceed human capacity, and chronic fatigue, physical overwork and even burn-out syndrome.

(d)Age distribution of children in the care of respondents

The majority of respondents, 73% (Figure 3) who participated in the study, care for children in the age group 10–18 years, which means that life-limiting illness can affect children’s attendance and school performance. Parents may experience frustration and worry about the need to manage and adapt the educational environment to suit their child’s specific needs. As children get older, they may become more aware of their illness and the impact on their lives. Parents may be involved in managing the emotional and psychological aspects of the illness, including anxiety, depression and other emotional reactions of their child.

(e)Distribution by type of pathology of children

Analyzing the pathological palette for which the children included in the study came to our center to receive special care, it is observed that most of them presented chronic infantile encephalopathy with spastic tetraparesis (26%); others presented malformations of the central nervous system and spinal cord (17%), serious oncological diseases (neoplasms or lymphomas) (15%) or genetic diseases (12%). A further percentage (9%) was attributed to children with hydrocephalus and plurimalformative syndrome, and 6% had congenital cardiac malformations (Figure 4).

Regardless of the child’s age, medical, social, psycho-emotional and spiritual needs are the same. These children require education, socialization and remedial therapy, and they need to be valued and understood despite the fact that most of them are chronically ill with progressive diseases, require complicated care and are prone to complications (cerebral palsy and multiple pathologies following brain or spinal cord injuries) and imminent premature death.

The majority of children (26%) cared for by the study participants have conditions characterized by complex medical needs and multiple disabilities, including the most aggressive, irreversible conditions, requiring complex medical services and involving the constant risk of complications. Access to palliative care services is vital because the family needs a break to recover and continue care and cannot meet the needs of treatment and response to other illness-related conditions. Central nervous system and spinal cord malformations (17%) may be diagnosed prenatally or at birth. Diagnosis often involves complex medical investigations, including advanced medical imaging, and may involve surgery early in the child’s life, which implies a long-term responsibility. The family must cope with the need to provide ongoing care, including administering medical treatments, therapy and managing daily needs.

Oncological diseases in children (15%) are a painful reality for affected families because of their seriousness. Palliative care becomes essential in these situations, given the emotional and physical impact on the child and family. Children can develop various types of cancer, the most common being leukemia, brain tumors, neuroblastoma, osteosarcoma and lymphomas. These conditions require different approaches to treatment and care, and intensive medical treatments such as chemotherapy, radiotherapy and surgery can cause significant side-effects (nausea, extreme fatigue and hair loss) that put the family at risk of giving up.

If the disease progresses and the care is long-term, parents face physical loss of the child. Preparing for and managing this loss can be an extraordinarily difficult emotional challenge. Providing emotional support, access to counselling resources and specialist assistance can play a crucial role in helping parents cope with this difficult challenge.

#### 3.1.2. Analysis of the Data Regarding the Concept of “Respite”—Parents’ Perspective

(a)The “respite” concept

When asked if they were aware of the concept/definition of the term “respite”, 68% of the respondents said “NO” and 32% said “YES”.

(b)Participating in the survey

When asked “What do you think are the main benefits when you receive respite services?”, two categories were identified. Category A referred to “time”—that is, time for the mother/primary caregiver to recharge; to devote to family or partner; to go to arts, sports and cultural events; to deal with duties she has not performed; for medical check-ups; to learn caregiving techniques; to seek guidance and support for stress or overload; and to attend support groups. All 34 respondents in our research ticked these benefits. This highlights the heavy burden that has accumulated during caregiving, highlighting that such a service would be like a “breath of fresh air” for them and their families. 

The other category, B, which none of the respondents chose to tick as a benefit, was, “it would help me to focus on my job and to keep it”, which means that the participants in this research do not have a job other than caring for their own child, assisting them in their moments of struggle with grief and giving them comfort and support.

Unlike other jobs where you spend 8 h/day, 5 days a week, these people work 24/7. When asked in the questionnaire about the weekly time spent with the child, all respondents in the study answered that they work more than 40 h/week.

(c)When asked how they were informed about the “respite’’ care service

Analyzing the answers to the questionnaires, the proportions of people who have heard about the specialized and respite services in our center from the various sources are similar. This can be observed in Figure 5. This shows that most of the institutions with which the organization collaborates inform the families concerned about the benefits of these types of care, and the fact that it is a complex specialized service that has a multidisciplinary team that is concerned with the care of the child and his family.

Despite the growing number of studies and policies at the international level related to improving support for parents who are the primary caregivers of a child diagnosed with a life-limiting illness [25,26], in Romania, adequate specialized palliative and respite care services remain problematic because there is no uniform development at the national level [27], and pediatric palliative care is still considered “the ash heap” of the medical system. 

#### 3.1.3. Analysis of the Data in Terms of the Coping Pattern Adopted by the Respondents to the Study

In order to cope with situations that arise during care, our respondents have developed certain coping strategies, which are actions, behaviors and thoughts that they use to cope with the situation. 

Coping methods were evaluated, and they revealed (Figure 6) the following situation: seeking help from friends in 12% of the study participants; positive thinking in 15% of the respondents; changing the way of perceiving the situation in 23% of the respondents; seeking information, another opinion, asking questions in about 32% of our study respondents; avoiding stressful situations, including contact with the hospital or other family members who did not prove to be supportive, in 18% of the study participants.

Analyzing the above data shows that 32% of respondents who use the coping method “seeking information, seeking another opinion, asking questions” admit that this strategy is relevant or effective for them in managing stress or difficult situations.

This coping method implies that people manage their emotions or problematic situations by seeking information, consulting other opinions or asking questions. This may suggest that they feel more comfortable or prepared to cope with stress by gaining additional knowledge, seeking alternative perspectives or clarifying issues through questioning.

### 3.2. Analysis of Focus Group Meetings

There were 12 female participants in the focus group meeting, other than those who responded to the questionnaire, who received respite palliative care services, and two of them did not receive respite palliative care services in the Children’s Palliative Care Centre, but only received palliative care services at home. The focus group participants were aged 32–42 years old, residing in both urban and rural areas. The focus group was led by two researchers involved in this study.

The focus group discussions led us to identify several themes that correspond to the difficult situations often faced by parents of children diagnosed with a chronic progressive/life-limiting illness, situations/conditions to which we will refer below: Confusion about the understanding of the term “respite”/“respiro”:


*“I didn’t know what it means…is there such a thing in Romania? without paying…is it for me? Or for the child?” (focus group participant).*


Financial difficulties are another category of difficult situations faced by parents of children diagnosed with life-threatening/life-limiting illnesses:


*“We can’t manage on our own…sometimes we feel like we are falling apart…and we are also very poor” (focus group participant).*


The need for time for activities other than those related to the child’s medical condition, socializing, time with other children, time for assessing their health:


*“For the first time in 15 years…I went with my husband and other children on a holiday…I have never been since the sick child was born” (focus group participant).*



*“I went to have my medical tests done…since the birth of my little one…8 years ago…I had no one to leave him with” (focus group participant).*


The low access to different forms of therapy for children (physiotherapy, psychological counselling) is also determined by the low number of specialists competent in pediatric palliative care and the low number of such services to work with children:


*“I know several parents who need help, in the same situation as mine, with a sick child…and their siblings need…to be listened to…if we had to pay a psychologist…we can’t afford it…we spend a lot of money on medicines…food…utilities” (focus group participant).*


Concern about the future, about the adult lives of children who will no longer be able to access pediatric palliative care services:


*“Now, while he is a child…I have help from the Light Association, but when he is an adult…where will he feel loved and cared for? It will be hard to find a place that gives us a rest from caring for him, day and night” (focus-group participant).*


All the focus group participants said that they had experienced problems in their relationship with their husband and that during the period when the child was in the respite center, they were able to renew their relationship with their partner and to pay more attention to him.

All focus group participants said they were able to spend more free time with other children on holidays or in extracurricular activities and were able to visit their sick parents.

For the parents, life before the child’s diagnosis was typical, with the everyday problems they faced posing no particular difficulties. Family life was seen as harmonious, with the birth of children being something the family wanted and prepared for. The child’s diagnosis led to a change in the lifestyle of the whole family, the most important consequence of which was a reduction in the amount of time devoted to relaxing, socializing and leisure activities.

It appears that for all of our survey respondents, families of children with a progressive/life-threatening chronic illness are giving up leisure activities that they were doing prior to the child’s diagnosis. Gradually, a phenomenon of social withdrawal takes place, outings with friends are reduced in frequency and their mental state deteriorates from day to day.

## 4. Discussion

Palliative care can be provided to children diagnosed with life-limiting illnesses from the time of diagnosis and can last for several years and continue until death [28]. The trajectory of life-limiting illnesses for children is often unpredictable, often bordering on death, making their care difficult to manage. This, therefore, explains why providing palliative care services for these children can be a great challenge, and flexibility on the part of specialists is needed to cope with changes in the disease trajectory [29]. 

The literature also indicates that access to palliative care services is a real challenge; accessibility and reduced availability of these services have been a barrier to meeting a person’s needs [30]. 

The current health care workforce situation (lack of health care specialists) and funding restrictions mean that there are no services available, and all of this influences the level of support that children and their families receive when they are faced with life-limiting and life-threatening conditions [31,32]. Also, responses to the “respite term” suggest that this term is not known and is not promoted, and that the Romanian medical-social system has not developed this service much. There are reasons why this service is not developed: it is not known how many children need specialized palliative care, there are not enough specialists working in such services and there are no financial resources to develop pediatric palliative care.

Our study led to several key findings, all suggesting that the respite program is beneficial. The beneficiaries of the program were parents of children diagnosed with life-limiting illnesses, with special care needs, and siblings. The most important benefit for the child’s parents was that it allowed them time to relax together. Parents were able to enjoy time off without having to constantly deal with the responsibility of caring for their child with special needs. 

Similarly, in a study by Kvarme et al. (2016), researchers found that parents of children with special care needs had limited time to evade caregiving responsibilities and were constantly exhausted. Greater access to respite care for parents of children diagnosed with life-limiting illnesses is one way in which the burden of caregiving can be alleviated [33].

Another study by Nishigaki, Yoneyama, Ishii and Kamibeppu (2016) on the benefits of respite identified that mothers who left their children in a respite center were anxious, fearful of having to part with their child and leave them in the care of strangers [34]. However, a key finding of this study suggests that respite care is not a “one size fits all”; instead, the respite care program must “fit” the needs of the family. This is also consistent with Whitmore’s (2016a) study, which concluded that the most important things for a caregiver to have adequate respite, and to reap the potential benefits of respite care, are to know the type, location, safety, duration, timingand frequency of the respite service, and to trust in the respite provider [35].

Access to a specialized ”respite” service is a limited resource, especially in Romania, and therefore, many families turn to other informal alternatives such as family and friends. Families in our study initially turned to relatives, neighbors and friends, but for a short period of time—a few hours—and never had the courage to leave the child for 10–14 days to have a successful leave with the other children [36].

In the NICE (End-of-life care for infants, children and young people with life-limiting conditions: planning and management) guideline (2016) [37], there are six gold standards for the provision of palliative care services for children at the end of life [38].

These standards should be met so that providers of palliative care for children provide a better quality of life for parents, siblings and children and young people receiving end-of-life care. The standards emphasize that families should: (1) be involved in the development of an advance care plan, (2) have a designated physician specializing in palliative care to coordinate their care, (3) be provided with information about emotional and psychological issues, including how to access them, (4) be cared for by a multidisciplinary team, (5) be offered support to families when their child is nearing the end of life and after their death, and (6) have 24 h access to specialist pediatric palliative care support when nearing the end of life and when being cared for even at home [38].

Several studies in the field of palliative care for children have identified that the most common unmet needs are the family’s need for “respite” [39,40,41,42,43], communication of staff involved in the child’s care due to lack of communication skills [43,44,45], coordination and organization of services [39,45,46] and emotional and psychological support [43,47,48].

Respite periods allow parents to rest, feel relieved of the responsibility of caring for their sick child for a few days, focus on their other children and their life as a couple, and break the social isolation. For siblings, the main benefits are that they can have their parents to themselves and share activities with them [21]. 

Participants in our study highlighted that respite care provided regularly over periods of time could enable them to better cope with the demands of caring for their children. This type of care for the child and family can take place at any stage of the disease course and should be seen as essential to the provision of quality care.

Parents/caregivers, as well as skilled caregivers, often observe that during the course of life-threatening illness, against the backdrop of patients’ immunosuppression, bacterial infectious complications arise that require treatments with newer-generation or escalating antibiotics (cephalosporins, colistin, carbapenems) [49,50]. Antibiotic therapies, although individualized, inevitably generate microbial resistance phenomena, the growth of which, worldwide, is life-threatening [49].

Parents of children diagnosed with life-limiting illnesses and complex medical needs can be overwhelmed by the ongoing care needs of their children. Caring for a child with special health care needs is often challenging, in many cases requiring specialized training. As a result, parents may struggle to find qualified caregivers capable of giving them a break from 24/7 care of their child. Respite care programs are designed to give caregivers a much-needed temporary break.

Understanding the interplay between formal support services, informal networks, and spiritual communities is crucial in comprehensively addressing the complex needs of families navigating the care of children with serious medical conditions. Members of the extended family (such as grandparents) may frequently step in to provide emotional, practical, and sometimes financial support to their families when a child is diagnosed with a life-limiting illness. They may assist with caregiving responsibilities, offer respite to parents, and provide a stable presence for the ill child and their siblings. 

In addition to the familial support, religious groups and spiritual communities can offer support to families facing life-limiting illnesses, such as comfort, hope, and guidance, helping individuals navigate the existential and emotional challenges that accompany serious medical conditions. Religious beliefs and practices can provide solace, meaning, and a sense of community to families grappling with uncertainty and grief. The matter of support networks in patients such as the ones studied is scarcely researched, especially in Romania. As such, future research in this area has the potential to not only enhance the quality of care provided to children with life-limiting illnesses but also to support the well-being of their families.

### Study Limitations

As the questionnaires have been administered, we have found that the wording of some questions was a little unclear to many respondents, often requiring clarification from us as researchers. Most of the caretakers involved in the study were not familiar with the terms addressed in the questionnaire questions, and some did not answer all the questions in the questionnaire. A pretest of the questionnaire was not conducted because parents of children with life-limiting illnesses were involved in the care of their children, making it difficult to participate in a pretest.

Another limitation is the absence of a question in the questionnaire about the involvement of other family members in the child’s care, particularly in relation to the child’s physical condition.

More detailed and in-depth information can be obtained through much more detailed techniques, including face-to-face interviews, telephone interviews or group interviews (Smith 1972, Beed 1985, Abramson 1990, Mishler 1991). These methods may cost more, but can provide richer information. The focus group research method does not have a precise definition: the term actually refers to a variety of techniques (Carey 1994: 226) [51], but the broadest definition could be the following: an interview conducted with a constructed group, which is focused on a certain topic and on a certain category of subjects, and the subjects can influence each other through their answers.

Lastly, it is important to note that, although information from the international literature was used and applied, this questionnaire is not validated, as there are no alternatives, for the time being, in Romania.

## 5. Conclusions

In our study, the majority of respondents (62%) are from rural areas, have limited access to healthcare, internet, information about specialist services, recovery therapies and support through support groups, which indicates that in rural areas, the burden of care for caretakers is high, and these people need support and respite to cope with the challenges of incurable illness. 

Meeting the needs of rural people requires good collaboration with primary care, where there is also provision of information and educational resources about the child’s illness and ways to manage care. In this way, they can become more independent in managing the complex needs of their children.

The respondents to our study who are in full physical and intellectual capacity often take on this responsibility alone, 24 h a day, in addition to household tasks or caring for other children; this situation has a significant social impact, and can lead to depression, irritability and anxiety due to sleepless nights, household chores that are often beyond human capacity, as well as chronic fatigue, physical overload and even burn-out syndrome. 

Recent medical and technological advances have meant survival for many children who would previously have died from their chronic illnesses. Now their lifespans have increased significantly, living with their families despite multiple complex medical needs. The use of respite services has been associated with a significant reduction in psycho-emotional distress by primary caregivers. 

Also, as a result of accessing respite palliative care services, it has been found that there is a strong link between the use of this type of service and a subsequent decrease in the number of hospital days in pediatric hospitals. 

As a result of the analysis of the questionnaire responses and focus group discussions, it was concluded that the benefits of the respite time that a specialist palliative care service can provide to the family can help in the following ways:Reducing stress and exhaustion: Caring for a sick child can be physically and emotionally exhausting. By providing a respite service, the family has the opportunity to relax and reduce overall stress and fatigue.Improving family relationships: By providing respite time, family members can spend quality time together or give attention to other family members, which can im-prove relationships and connections between family members.Regaining personal autonomy: Respite allows family members to regain a degree of personal autonomy, allowing them to take care of themselves and engage in activi-ties they value.Source of social support: Respite can also provide opportunities to connect with other community members or support groups. This can help increase social support and share experiences with other parents facing similar situations.

This research may not be transferable between countries due to differences in health systems and respite care provision; however, although pediatric palliative care in Romania is in its infancy, this may present an opportunity for international collaboration.

Respite services must be provided in an appropriate format and location acceptable to the family, tailored to meet the individual needs of the family.

Future research should consider the impact of severity of diagnosis, cultural and geographical factors and frequent deficits in equity of respite provision. Future research should also take into consideration the comparison of this type of program with other similar ones that deal with life-threatening illnesses.

## Figures and Tables

**Figure 1 healthcare-12-00748-f001:**
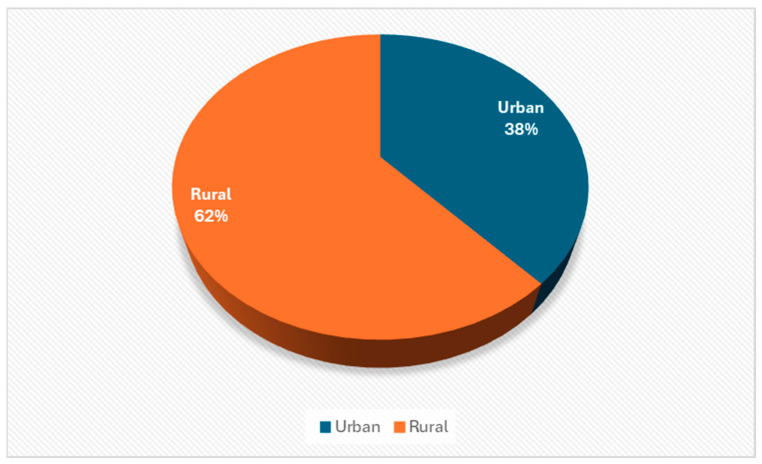
Distribution of respondents by background.

**Figure 2 healthcare-12-00748-f002:**
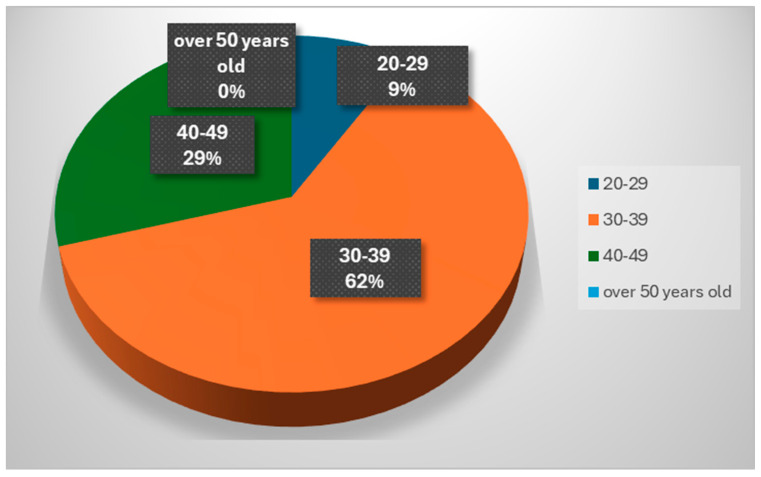
Distribution of respondents by age group.

**Figure 3 healthcare-12-00748-f003:**
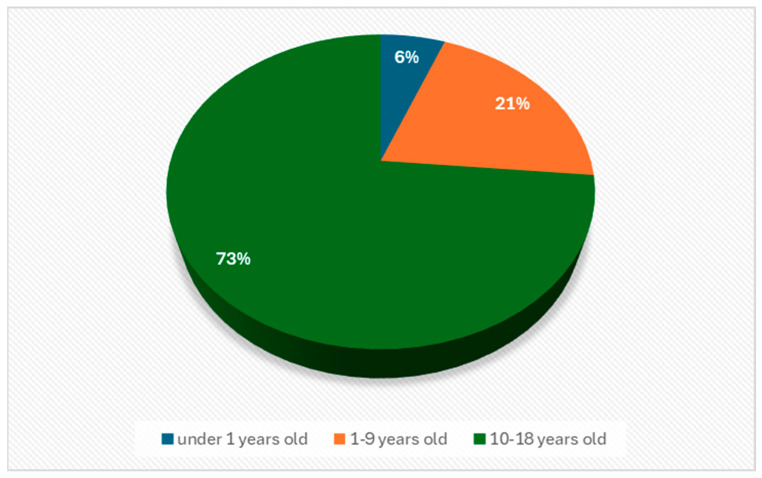
Distribution of children’s age.

**Figure 4 healthcare-12-00748-f004:**
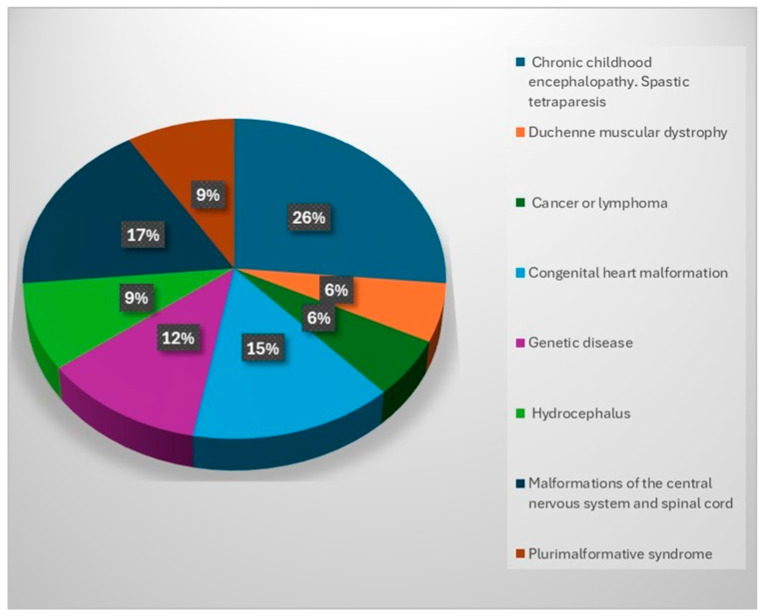
Distribution by type of pathology of children.

**Figure 5 healthcare-12-00748-f005:**
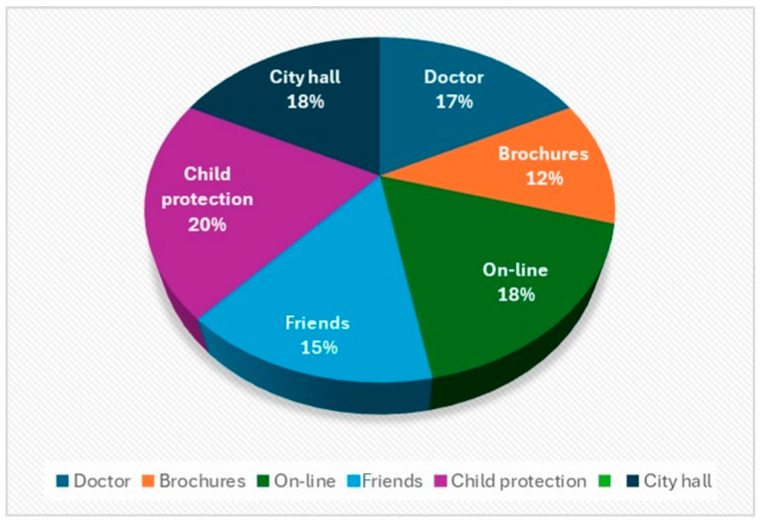
How respondents found out about the “respite” service.

**Figure 6 healthcare-12-00748-f006:**
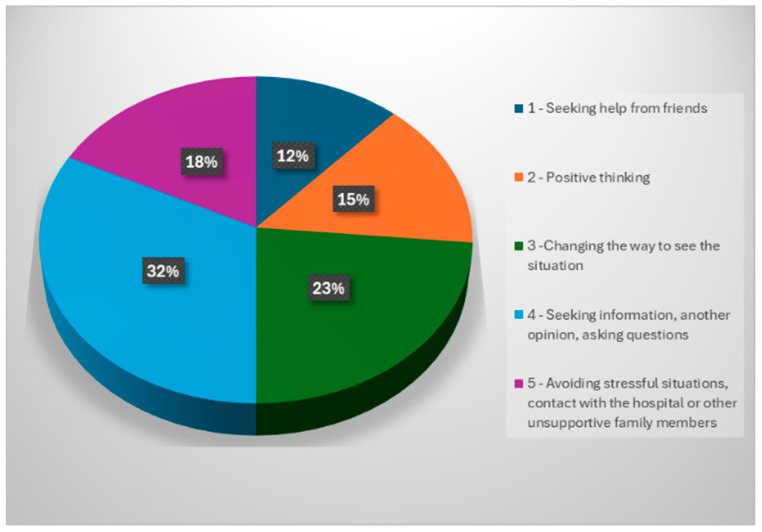
Coping patterns adopted by respondents.

## Data Availability

All data are available from the corresponding author.

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
