# Peer review of "Benefits of Respite Services on the Psycho-Emotional State of Families of Children Admitted to Hospice Palliative Care Unit: Preliminary Study on Parents’ Perceptions"

_healthcare, 2024, doi:10.3390/healthcare12070748_

Round 1

Reviewer 1 Report

Comments and Suggestions for Authors

Dear authors, it is a pleasure to review your text. I would like to make a series of considerations in this regard, in order to help you improve the quality of the text:

Introduction.

There is a good contextualization of the subject, although the explanation of the respite units could be improved (when did they originate, in what context, where are they used,...).

Material and methods.

The objective should be at the end of the introduction.

The psychometric properties of the questionnaires used are not specified, nor in what contexts they have been used previously.

There is no mention of the design of the study, the recruitment of the sample or the selection criteria.

The methods used in the focus groups are not specified, nor how the information obtained was transcribed, or which categories were contemplated in this type of design. The quantitative part of the study does not mention the variables either.

Author Response

Thank you for the comments and suggestions. Please find the responses attached. 

Reviewer 2 Report

Comments and Suggestions for Authors

Thank you very much for the opportunity to review this article. The article is very interesting and the topic studied is of great importance to parents caring for children with life-limiting illness.  However, I have a few questions:

a) in terms of the objective of the study, I was confused, does it intend to evaluate the parents' perceptions and the benefits it cna bring in improving their phycho-emotional state? It seems to me that what you wanted was to describe the parents' perceptions and evaluate the respite program. To clarify this point.  

b) In the introduction, you fail to understand how the program works. How long has this program existed? Who has had access to it? How many have had access? It is necessary to describe the program so that the reader can understand and frame the need for the study.

c) Materials and Methods - There is a lack of description of the instruments used, namely the questions in the questionnaires. I suggest putting the questions in a table. On the other hand, you mention that the questionnaires contain 26 multiple-choice questions.

d) Also with regard to the focus group, it is not clear who the experts are to carry out the analysis;

e) In the results, there is already some discussion. I think it's important to separate the presentation of the data and the analysis in points 3.1.2. 

e) I don't understand how they carry out the content analysis of the focus groups. 

d) In the analysis, they hypothesized that they had countries that didn't want to join the program because they couldn't leave their children at any time.

e) In the limitations of the study, they mention that the questions may not have been clear. They did not pre-test the questionnaire. In the methods section, this item needs to be clarified, otherwise it is not a limitation.

f) The conclusions go beyond the results of the study. They reach conclusions that the study did not. 

In short, I think you need to review your methodological options. You have some very interesting material here, but you need to be rigorous in your presentation. Once again, I don't think you should evaluate the benefits of the program, but rather the program, otherwise you're already biased, because you're already going in with an apriori view of the benefit.

Author Response

(The authors gave the same response as above.)

Reviewer 3 Report

Comments and Suggestions for Authors

On 1st read, this seemed an interesting report on a program familiar elsewhere but a novelty for rural Romania. There, a shortage of funds, services and professionals trained to work in this field of care means that any study of the respite care program would be a help. However, on further reflection, this essay seemed less a true research study, and more of an advertisement for their program. Yes, there are benefits, and parents found time away from continuous care for their sick child helpful. But what new is there to this? In a setting where resources are very limited, how does this service measure up against other programs with different aims that deal with life-threatening illnesses? The essay does not address this kind of question, nor look at why some traditional resources for care-givers aren't there. "It takes a village," or at least extended families, to share the responsibilities- and that is clearly not the case here. What happened? Where are grandparents? What about the church- does it have any support resources for these over-burdened families? Or- as it seems from this essay- is this once again a case of medical advances that keep persons alive longer, but without the social network to support them successfully? I think the essay would benefit if these questions were included, and if it sounded less like a marketing of the authors' program.

Comments on the Quality of English Language

Only minor problems. The content of the essay is what bothers me.

Author Response

(The authors gave the same response as above.)

Round 2

Reviewer 1 Report

Comments and Suggestions for Authors

Dear authors, following the requested changes, the text has been significantly improved.

Reviewer 2 Report

Comments and Suggestions for Authors

I think the authors have made an effort to improve the article following the recommendations.

Reviewer 3 Report

Comments and Suggestions for Authors

This is improved by the inclusion of context: Romania's healthcare system, the lack of many supportive systems and the overall impoverished quality of care available for these rural, stressed-out families. This information makes the present study seem more important, as a report on one now-available program. The study does not present anything new and surprising about the benefits of respite care, but now shows how it can help in one particular country.